



# Analytical and Numerical Solutions of Radially Symmetric Aquifer Thermal Energy Storage Problems

Zerihun K. Birhanu[1,3], Nils-Otto Kitterød[2], Harald Krogstad[3], and Anne Kværnø[3]

[1]School of Mathematical and Statistical Sciences, Hawassa University, Ethiopia
[2]Faculty of Environmental Sciences and Natural Resource Management, Norwegian University of Life Sciences, Norway
[3]Department of Mathematical Sciences, Norwegian University of Science and Technology, Norway

*Correspondence to:* zerie96@gmail.com

**Abstract.** The paper discusses analytical and numerical radial solutions of the differential equations for heat transport in water-saturated porous media. In particular, a similarity solution is obtained for a 2D-horizontal confined aquifer with constant radial flow. Numerical solutions are derived using a high-resolution Lagrangian approach suppressing spurious oscillations and artificial dispersion. There is a good correspondence between numerical and analytical solutions.

5  The primary purpose of the investigation has been to calculate the recovery factor of an Aquifer Thermal Energy Storage (ATES) system with a cyclic repetition of injection and pumping. Solutions covering both instantaneous and delayed heat transfer between fluid and solid, as well as time varying water flow, are derived and applied to a one-well test case.

In hydrological terms, these solutions are relevant for a wide range of problems where groundwater reservoirs are utilized for extraction and storage (viz. irrigation; water supply; geothermal extraction).

## 1 Introduction

Heat transfer in porous media has received considerable attention and is the topic of a number of investigations during the last decade (Bejan and Kraus, 2003). A driving force for research on this subject is engineering applications, such as geothermal systems (Ganguly and Kumar, 2014), heat exchangers (Diao et al., 2004), thermal insulation (Birhanu et al., 2015a; Kim et al., 2010), and safety issues regarding storage of nuclear waste (Sun et al., 2010).

In addition to equations for the fluid flow, the mathematical model of heat transfer in porous media is given by second-order partial differential equations for heat energy conservation and flow in the model domain. Two kinds of models can be applied to investigate thermal characteristics of conduction and advection within a porous medium, namely, a thermal equilibrium model and a thermal nonequilibrium model (Yang et al., 2011). The difference between the two models is the thermal coupling between the liquid and the solid phase. For the equilibrium model the coupling is modeled as an instantaneous heat transfer. This assumption is close to the reality for homogeneous aquifers with solid particles of minor size (diameter $d_p < 1$ mm). For the non-equilibrium model there is a time delay attached to the heat transfer between the two phases. In the literature of transport in porous media, this model is usually called a double porosity model which may also be expanded to a dual permeability



model (Stadler et al., 2012).

Researchers have highlighted different analytical and numerical methods to find the solution for this model based on different physical phenomena. Bear and Cheng (2008) presented the solution of heat transfer in porous media for 1D by using a similarity solution method. A two-dimensional numerical model for heat transport in heterogeneous porous aquifers has recently been presented by Ganguly et al. (2014). The model is validated analytically for a transient heat transport case with hot water injection.

Aquifer Thermal Energy Storage (ATES) is an example of technology where subsurface storage and transport of heat is used to save energy. ATES systems may utilize inter-seasonal heat storage, which means storage of excess energy from summer for being used in winter time for heating purposes. For cooling purposes low temperature water is extracted from cold wells and heated water is injected into hot water wells (Birhanu et al., 2015a). Thus, ATES installations actively store cooled and heated groundwater in the ground from respective heating and cooling mode cycles (Dickinson et al., 2009). An ATES system involves both the flow of water and heat transport. In order to predict the performance and efficiency of an ATES system, one possibility is to run detailed numerical simulations, and researchers have long highlighted numerical modelling for analysis and optimization of ATES systems (Lee, 2011). In some cases, models of injection and pumping wells may be simplified to a classical dipole geometry. In this case, the flow field may be simplified to axis symmetry where the flow velocities are governed by the injection/pumping rate and the aquifer porosity. In the present study, we take advantage of a simplified flow field and solve the transport equation by simple analytical and numerical methods to evaluate the energy efficiency of an idealized ATES system. Numerical solutions of transport problems are usually affected by artifacts, and because all mathematical models are simplifications of reality, boundary conditions should be specified with great precaution. Birhanu et al. (2015b) showed that the most evident boundary condition of temperature at the top of an unconfined aquifer gave unphysical energy efficiency for simulation experiments of a real ATES system. In this study we use analytical solutions to understand the quality of numerical simulations by doing (simple) numerical experiments. One such experiment is an idealized ATES production sequence consisting of repeated injection and pumping of hot water in a confined aquifer. The performance of the alternative solutions is quantified by a recovery (or efficiency) factor. The results are presented in Section 5.

## 2  Mechanisms and Equations

A thermo-hydraulic analysis requires calculation of simultaneous water and heat transport in an aquifer consisting of a solid porous medium ($s$) with pores filled with water ($w$). The water flow depends on properties of the water as well as the solid, and the gradient of the hydraulic head, as stated in Darcy's law (Molson et al., 1992; Nield and Bejan, 2013):

$$\mathbf{q} = -\frac{\rho g \mathbf{k}}{\mu} \nabla \left( \frac{p}{\rho g} + z \right) = -\mathbf{K} \nabla \phi. \tag{1}$$





Here, $\mathbf{q}$ is the specific discharge or the Darcy velocity, $\mathbf{k}$ the intrinsic permeability tensor, $z$ the elevation of the piezometric head relative to a datum level, $p$ the fluid's pressure, $\rho$ water mass density, $g$ the acceleration of gravity, and $\mu$ the dynamic viscosity of water. Furthermore, $\mathbf{K} = \rho g \mathbf{k} / \mu$ is the hydraulic conductivity, and $\phi = p/\rho g + z$ is the hydraulic head. The volume average velocity differs from the velocity of the water in the pores, the so-called seepage velocity, $\mathbf{u} = \mathbf{q}/n$, where $n$ is the (effective)

porosity (Kangas and Lund, 1994). The water density and, more pronounced, viscosity vary with temperature. However, we shall here assume that the flow field represented by $\mathbf{q}$ remains independent of the temperature changes. Assuming that the solid matrix as well as the water are incompressible, mass conservation combined with Darcy's law leads to the Poisson equation for the hydraulic head,

$$\nabla \cdot [-\mathbf{K}\nabla\phi] = Q_w(t), \tag{2}$$

where $Q_w(t)$ is a source/sink term.

The heat energy content per aquifer volume unit may be written

$$\rho_w c_w T_w n + \rho_s c_s T_s (1-n), \tag{3}$$

where $c$ is the specific heat, and subscripts $w$ and $s$ refer to water and solid. At a local temperature equilibrium where, $T_w = T_s = T$, the heat content may be expressed as $(\rho c)_m T$, where

$$(\rho c)_m = \rho_w c_w n + \rho_s c_s (1-n), \tag{4}$$

see Kangas and Lund (1994) and Hecht-Méndez et al. (2010). In the following we will use the convention $\rho_m c_m$ for $(\rho c)_m$. The water flow causes advection of heat,

$$\mathbf{q}_c = (\rho_w c_w T_w)\mathbf{q}, \tag{5}$$

whereas conduction/diffusion of heat takes place both in the solid and the liquid,

$$\mathbf{q}_w = -n\lambda_w \nabla T_w, \tag{6}$$
$$\mathbf{q}_s = -(1-n)\lambda_s \nabla T_s, \tag{7}$$

and $\lambda_{w,s}$ are the heat diffusion coefficients. If the two media are at a local thermal equilibrium, the volume average diffusive heat flux may be expressed by

$$\mathbf{q}_T = -\lambda_m \nabla T, \tag{8}$$

where $\lambda_m$ is a bulk aquifer heat diffusion coefficient,

$$\lambda_m = n\lambda_w + (1-n)\lambda_s, \tag{9}$$





see Kangas and Lund (1994) and Nield and Bejan (2013). Other expressions for $\lambda_m$, e.g. porosity-weighted geometric and harmonic means are also discussed in the literature (Nield and Bejan, 2013). In addition, the heterogeneity of the pores induces a certain amount of thermal dispersion, parametrized, in its simplest form as

$$\mathbf{q}_d = \rho_m c_m \hat{\alpha} |\mathbf{q}| \nabla T. \tag{10}$$

Here $\hat{\alpha}$ is the thermal dispersivity length, and the total diffusion flux becomes $\mathbf{q}_T + \mathbf{q}_d$ (Bakr et al., 2013).

If a thermally insulated aquifer with initial temperature $T_w \neq T_s$ and $\mathbf{q} = 0$, energy conservation implies that the system attains an equilibrium temperature $T_m$ equal to the weighted mean

$$T_m = \frac{\rho_w c_w n}{\rho_m c_m} T_w + \frac{\rho_s c_s (1-n)}{\rho_m c_m} T_s. \tag{11}$$

How fast the temperature equilibrium between water and solids is reached depends on the efficiency of the energy exchange between the two media. It turns out to be reasonable to express the heat exchange per time and volume unit as

$$P = h(T_w - T_s), \tag{12}$$

where $h$ is a heat transfer coefficient (Nield and Bejan, 2013; Kreith et al., 2010). The coefficient varies with temperature and flow, in particular for large flows. Following the discussion in Nield and Bejan (2013), $h$ may be expressed as $h = a_{ws} h_v$, where $a_{ws} = 6(1-n)/d_p$ is the surface area of the water/solid interface per volume unit. For low Reynolds numbers, $h_v$ may be expressed as

$$h_v = \frac{5\lambda_H}{2d_p}, \quad \frac{1}{\lambda_H} = \frac{1}{2}\left(\frac{1}{\lambda_w} + \frac{1}{\lambda_s}\right). \tag{13}$$

Here, $d_p$ is the size of the grains making up the solid. The expression for $h$ then becomes

$$h = 15\lambda_H (1-n) d_p^{-2}. \tag{14}$$

A rough estimate of the time scale $\Delta t$ towards thermal equilibrium may be obtained from the energy exchange per time unit at the start of the heating, $P = h(T_w - T_s)$, compared to the required amount of energy to be transferred, $E = (1-n)\rho_s c_s (T_w - T_s)$:

$$\Delta t = \frac{E}{P} = \frac{1}{15} \frac{\rho_s c_s}{\lambda_H} d_p^2. \tag{15}$$

The time scale is thus only dependent on basic material constants and the grain size. With typical values for rock, we obtain

$$\Delta t\,[\text{s}] \approx 0.15 \times (d_p\,[\text{mm}])^2. \tag{16}$$

A similar time scale may actually be derived from the heating of spheres discussed in Gockenbach and Schmidtke (2009).

For an elemental aquifer volume $R$ with boundary $\partial R$, the solid's integral conservation law reads

$$\frac{d}{dt}\int_R (1-n)\rho_s c_s T_s dV + \int_{\partial R} (1-n)(-\lambda_s \nabla T_s) \cdot \hat{\mathbf{n}} d\sigma = \int_R h(T_w - T_s) dV. \tag{17}$$





Note that since the solid is not moving in this case, there is no advective heat for the solid phase. Similarly, the integral conservation form for heat in the water is

$$\frac{d}{dt}\int_R n\rho_w c_w T_w dV + \int_{\partial R}[\rho c_w T_w \mathbf{q} - n\lambda_w \nabla T_w]\cdot\hat{\mathbf{n}}d\sigma = -\int_R h(T_w - T_s)dV. \tag{18}$$

The differential forms of the conservation laws with the assumptions above become

$$(1-n)\frac{\partial}{\partial t}\left(\rho_s c_s T_s\right) - (1-n)\nabla\cdot\left(\lambda_s\nabla T_s\right) = h\left(T_w - T_s\right), \tag{19}$$

$$n\frac{\partial}{\partial t}\left(\rho_w c_w T_w\right) + \nabla\cdot\left(\rho_w c_w T_w\mathbf{q}\right) - n\nabla\cdot\left(\lambda_w\nabla T_w\right) = -h\left(T_w - T_s\right). \tag{20}$$

We observe that when $T_w$ is kept constant and diffusion is neglected, the natural time scale (inverse rate of change) in Eq. (19) is essentially $\Delta t$. For $d_p$ less than about a millimetre, the thermal equilibrium is virtually spontaneous and we may assume that $T_w$ and $T_s$ are equal.

For the case where $T = T_w = T_s$, we obtain by adding Eq. (17–18),

$$\frac{d}{dt}\int_R c_m\rho_m T dV + \int_{\partial R}[\rho_w c_w T\mathbf{q} - \lambda_m\nabla T]\cdot\hat{\mathbf{n}}d\sigma = 0, \tag{21}$$

and the corresponding differential form,

$$\frac{\partial\left(c_m\rho_m T\right)}{\partial t} + \nabla\cdot\left(c_w\rho_w T\mathbf{q}\right) - \nabla\cdot\left(\lambda_m\nabla T\right) = 0. \tag{22}$$

If the parameters like $c,\rho,\lambda$ and the flow $\mathbf{q}$ are assumed to be independent of $T$, then dividing through with $c_m\rho_m$ in Eq. (22) leads to

$$\frac{\partial T}{\partial t} + \nabla\cdot(\boldsymbol{\kappa}T) - \lambda\nabla^2 T = 0, \qquad \boldsymbol{\kappa} = \frac{\rho_w c_w}{\rho_m c_m}\mathbf{q}, \quad \lambda = \frac{\lambda_m}{\rho_m c_m}. \tag{23}$$

In our model, the flow $\mathbf{q}$ is caused by water injected or pumped with a discharge rate $Q(t)$ from a well located at the origin. When $Q(t) > 0$, water is injected from the well into the aquifer, causing a flow away from the well. During pumping, $Q(t) < 0$ and the flow is directed towards the well. Utilizing symmetric geometry of the aquifer near the well, the flow is $\mathbf{q} = q_d\mathbf{i}_r$ in which the discharge velocity $q_d = \frac{Q_d}{r^{d-1}}$ where

$$Q_d = \begin{cases} \frac{Q}{HW} & \text{for } d = 1 \text{ (linear flow)}, \\ \frac{Q}{2\pi H} & \text{for } d = 2 \text{ (radial flow)}, \\ \frac{Q}{4\pi} & \text{for } d = 3 \text{ (spherical flow)}. \end{cases} \tag{24}$$

The width $W$ and height $H$ are constants characteristic for the aquifer. In this case, Eq. (23) becomes

$$\frac{\partial T}{\partial t} + \frac{\kappa_d}{r^{d-1}}\frac{\partial T}{\partial r} = \frac{\lambda}{r^{d-1}}\frac{\partial}{\partial r}\left(r^{d-1}\frac{\partial T}{\partial r}\right), \qquad \kappa_d = \frac{\rho_w c_w}{\rho_m c_m}Q_d. \tag{25}$$

Using the same symmetry considerations in the nonequilibrium case Eq.(19–20) become

$$\frac{\partial T_s}{\partial t} = \frac{\lambda_s}{\rho_s c_s}\frac{1}{r^{d-1}}\frac{\partial}{\partial r}\left(r^{d-1}\frac{\partial T_s}{\partial r}\right) + \frac{h}{(1-n)\rho_s c_s}(T_w - T_s), \tag{26}$$

$$\frac{\partial T_w}{\partial t} + \frac{Q_d}{n}\frac{1}{r^{d-1}}\frac{\partial T_w}{\partial r} = \frac{\lambda_w}{\rho_w c_w}\frac{1}{r^{d-1}}\frac{\partial}{\partial r}\left(r^{d-1}\frac{\partial T_w}{\partial r}\right) - \frac{h}{n\rho_w c_w}(T_w - T_s). \tag{27}$$





## 3 Analytical Solutions

Analytical solutions express the relation between the principal variables involved directly. This provides basic insight to the problem without any further numerical evaluation. Besides this intuitive conceptual advantage, classical applications of analytical solutions are practical parameter estimation and sensitivity analysis. In practical hydrology physical parameters like permeability or storage capacity, are not always accessible for direct measurements. Response functions (viz. hydraulic head; temperature) on the other hand, are usually more easy to monitor. In such cases, the physical parameters are estimated by solving the inverse problem. Before the advent of computer technology, this was done by dimensionless solutions of the analytical expression, which provided tables or so-called type-curves. The inverse problem was solved by curve fitting of the empirical data to the analytical type curve. Today, analytical solutions are applied in similar ways, but the curve fitting is substituted by numerical perturbation of the involved variables. Sensitivity analysis is another example where application of analytical solutions is convenient. After estimation of optimal parameters, the relative impact of the uncertainties might be evaluated by simple numerical perturbation of the involved parameters. Here, in this context, the motivation for using analytical expressions of the transport equation is due to the numerical challenge of solving the transport equation if advective flow is dominant. In such cases, numerical solutions are prone to numerical dispersion. Even though analytical solutions simplify real transport, numerical artifacts do not affect the solutions. Therefore, by using the same simplified velocity field for both the analytical and the numerical solutions, the performance of the numerical algorithm can be evaluated directly by using the analytical solutions as benchmarks.

We shall consider the formation of a hot water plume in a local thermal equilibrium aquifer generated by a constant hot water source at the origin. Consider Eq. (25) for convenience normalized such that $\kappa_d = 1$ and with initial and boundary conditions,

$$T(r,0) = 0, \qquad T(0,t) = 1, \qquad \lim_{t \to \infty} T(r,t) = 0, \qquad r,t > 0. \tag{28}$$

If the diffusion term is negligible, Eq. (25) becomes a simple hyperbolic equation which, for any initial temperature distribution $T(r,0) = f(r)$, has the solution $T(r,t) = f\left(r^d - d\kappa_d t\right)$. In particular, for the boundary conditions in Eq. (28), the hyperbolic solution is the moving front $T(r,t) = H_c\left(r^d - d\kappa_d t\right)$ where $H_c(x)$ is the complementary Heaviside function ($= 1$ for $x \leq 0$, $= 0$ for $x > 0$). Since $\rho_w c_w$ is typically about twice as large as $\rho_s c_s$, the ratio $\frac{\rho_w c_w}{\rho_m c_m}$ depends on the porosity $n$ and varies between 1 and 2. The temperature front thus moves significantly faster than the discharge velocity $q$, but more slowly than the average seepage velocity, $u = q/n$.

### The 1-dimensional case

When $d = 1$ Eq. (25) becomes

$$\frac{\partial T}{\partial t} + \kappa_1 \frac{\partial T}{\partial x} = \lambda \frac{\partial^2 T}{\partial x^2}, \tag{29}$$




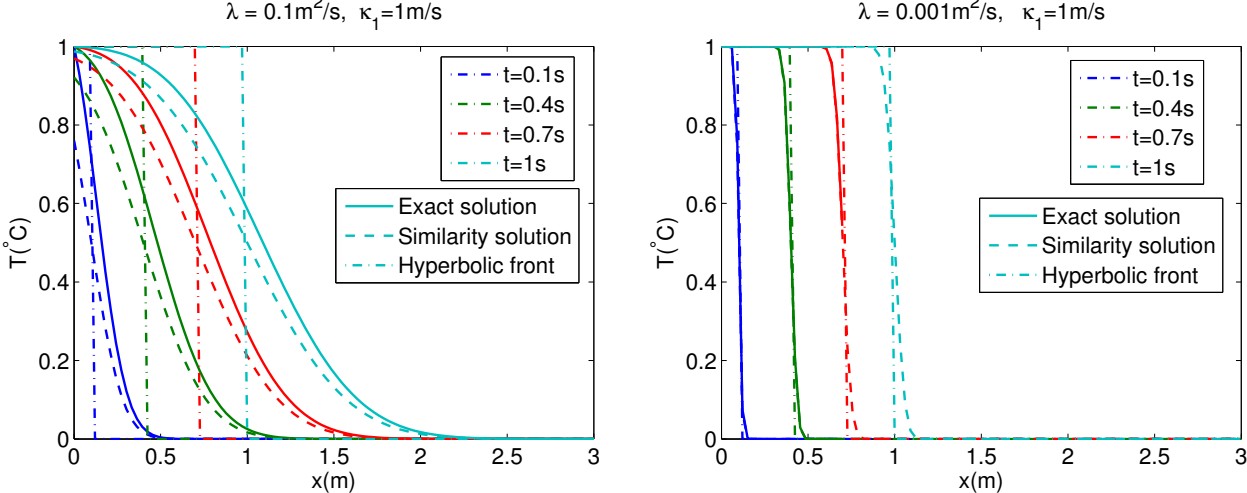

**Figure 1.** The 1-D similarity solution Eq. (31) and the exact solution Eq. (32) compared with the hyperbolic front for two different values of the diffusion coefficient $\lambda$.

where we choose the more standard spatial variable $x$ rather than $r$. In this case, a well-known similarity variable is $\eta = (x - \kappa_1 t)/\sqrt{\lambda t}$, which, inserted into the equation results in

$$2\frac{d^2 T}{d\eta^2} + \eta \frac{dT}{d\eta} = 0, \tag{30}$$

with the general solution $T(\eta) = C_1 \mathrm{erf}(\eta/2) + C_2$. However, no solution from this collection satisfies the boundary condition
$T(0,t) = 1$. Nevertheless, the similarity solution of the closely related problem satisfying the initial values $T(x,0) = H_c(x)$, $x \in \mathbb{R}$ is a good approximation:

$$T(x,t) = \frac{1}{2}\mathrm{erfc}\left(\frac{x - \kappa_1 t}{2\sqrt{\lambda t}}\right). \tag{31}$$

A modification of this solution, satisfying all conditions in Eq. (28) exactly has been derived by Ogata and Banks (1961); see also Bear and Cheng (2008), Eq. 6.4.30:

$$T(x,t) = \frac{1}{2}\left(\mathrm{erfc}\left(\frac{x - \kappa_1 t}{2\sqrt{\lambda t}}\right) + \exp\left(\frac{\kappa_1 x}{\lambda}\right)\mathrm{erfc}\left(\frac{x + \kappa_1 t}{2\sqrt{\lambda t}}\right)\right). \tag{32}$$

The similarity solution Eq. (31) and the exact solution Eq. (32) are presented in Fig. 1 together with the hyperbolic front solution $T(x,t) = H_c(x - \kappa_1 t)$. As expected, the similarity solution in Eq. (31) does not satisfy the boundary conditions at $x = 0$. Still, as $\lambda$ tends to 0, the solution approaches the hyperbolic front, and Eq. (31) becomes a very good approximation.

**The 2-dimensional radial symmetric case**

For a 2-dimensional problem, assuming radial symmetry, Eq. (25) becomes

$$\frac{\partial T}{\partial t} + \frac{\kappa_2}{r}\frac{\partial T}{\partial r} = \frac{\lambda}{r}\frac{\partial}{\partial r}\left(r\frac{\partial T}{\partial r}\right), \tag{33}$$





which may be rewritten as

$$\frac{\partial T}{\partial t} + \frac{\kappa_2 - \lambda}{r}\frac{\partial T}{\partial r} = \lambda\frac{\partial^2 T}{\partial r^2}. \tag{34}$$

Again, it turns out that assuming the similarity variable $\eta = r/\sqrt{\lambda t}$, we obtain an equation

$$\frac{d^2 T}{d\eta^2} = \left(\frac{\alpha - 1}{\eta} - \frac{\eta}{2}\right)\frac{dT}{d\eta}, \qquad \alpha = \frac{\kappa_2}{\lambda}, \tag{35}$$

with general solution

$$T(\eta) = C_1\int\limits_0^\eta s^{\alpha-1}e^{-s^2/4}ds + C_2. \tag{36}$$

The solutions may be written in terms of the incomplete $\Gamma$-function, defined as

$$\gamma(x,a) = \int\limits_0^x t^{a-1}e^{-t}dt, \qquad \Gamma(a) = \gamma(\infty,a). \tag{37}$$

The radial 2D similarity solution becomes

$$T(\eta) = 1 - \frac{\gamma\left(\frac{\eta^2}{4},\frac{\alpha}{2}\right)}{\Gamma\left(\frac{\alpha}{2}\right)}, \tag{38}$$

or

$$T(r,t) = 1 - \frac{\gamma\left(\frac{r^2}{4\lambda t},\frac{\alpha}{2}\right)}{\Gamma\left(\frac{\alpha}{2}\right)}, \tag{39}$$

and the solution is shown is Fig. 2.

The spherical symmetry 3D-case is easily seen to have intrinsic scales for $r$ and $t$ involving $\kappa_3$ and $\lambda$, and no simple similarity solution exists. The scaled equation may however be transformed to a 1D heat equation with a space dependent diffusion coefficient (Philip, 1994). Actually, the numerical algorithm below applies a similar transformation.

## 4 Computational procedure

We shall now consider a numerical algorithm for solving Eq. (19–20), or the equilibrium model Eq. (23) in the case of symmetric geometry. It turns out that the numerical solution of these problems is nontrivial. They are typically advection dominated, and we have already seen in the previous section that the temperature profile is a sharp front moving away from the source. In the radial and spherical case, the flow becomes very large close to the origin, leading to an almost hyperbolic equation in this region. Advection dominated problems are notoriously difficult to solve numerically. Popular schemes, like central differencing schemes result in unstable or spurious oscillatory solutions. Upwind discretization for the advection term avoids oscillations, but does create artificial diffusion, leading to a smoothed temperature front when applied on a coarse grid. Several other methods have been proposed and discussed in the literature, see e.g. Strikwerda (2004) and LeVeque (1992).





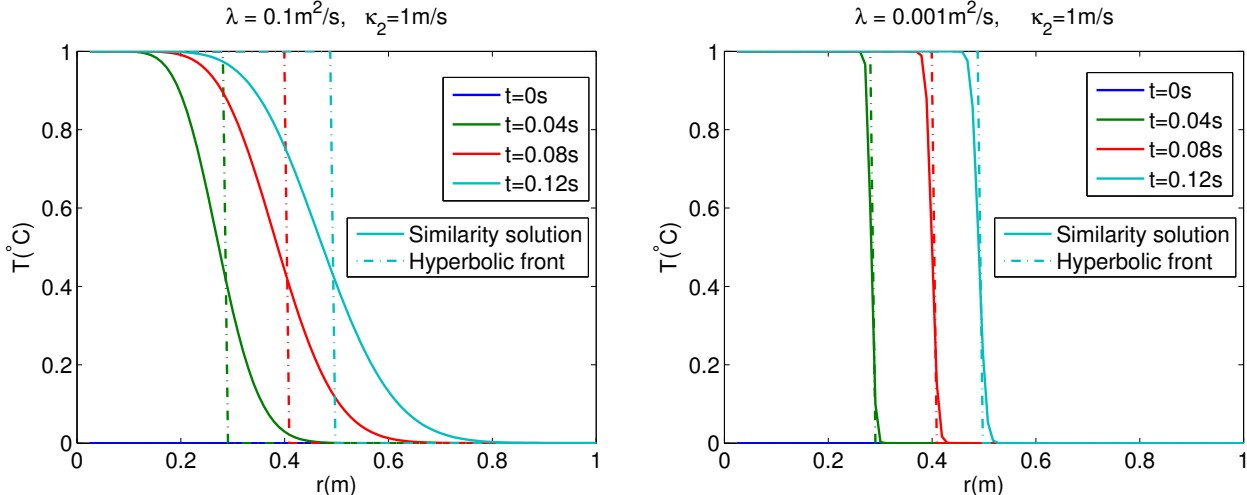

**Figure 2.** The exact similarity solution Eq. (39) compared with the hyperbolic front for two different values of the diffusion coefficient $\lambda$ in the 2-dimensional radial symmetric case.

We propose a computational procedure utilizing the special structure of the Eq. (25) or Eq. (26–27). The procedure is composed by the well-known Lagrangian approach combined with a coordinate transformation. The idea will first be explained for the equilibrium case Eq. (25).

Before discussing the numerical scheme, it is convenient to say something about the scaling of the problem. Consider a time

scale $\mathcal{T}$ for the time $t$, and $\mathcal{R}$ for the space variable $r$. A reasonable relation between the two scales is $\mathcal{R}^d = \bar{\kappa}_d \mathcal{T}$, $\mathcal{R}$ is the distance a hyperbolic temperature front moves in time $\mathcal{T}$ for a constant $\kappa_d = \bar{\kappa}_d$. The temperature $T$ will typically be scaled with the temperature of the injected water. With these scales we get the dimensionless equation

$$\frac{\partial T}{\partial t} + v(t)\frac{1}{r^{d-1}}\frac{\partial T}{\partial r} = \frac{\hat{\lambda}}{r^{d-1}}\frac{\partial}{\partial r}\left(r^{d-1}\frac{\partial T}{\partial r}\right) \qquad r > 0, \quad t > 0, \tag{40}$$

where $v(t) = \kappa_d(t)/\bar{\kappa}_d$, $\hat{\lambda} = \lambda\mathcal{T}/\mathcal{R}^2$. As a curiosity, notice that in the case of a constant $\kappa_d$, by choosing $\mathcal{R}$ and $\mathcal{T}$ such

that $\kappa_d\mathcal{T}/\mathcal{R}^d = \lambda\mathcal{T}/\mathcal{R}^2 = 1$, the parameters are completely absorbed, in the sense that also $\hat{\lambda}$ becomes equal to one. This is possible for $d = 1$ and $d = 3$, but not for $d = 2$. However, we will not pursue this curious issue any further here.

To handle the singularities in origin in the radial and spherical cases, we introduce the transformation $s = r^d/d$ so that $\partial s = r^{d-1}\partial r$, valid for all dimensions $d$. In this case, Eq. (40) becomes

$$\frac{\partial T}{\partial t} + v(t)\frac{\partial T}{\partial s} = \frac{\partial}{\partial s}\left(\hat{\lambda}a(s)\frac{\partial T}{\partial s}\right), \qquad a(s) = (d \cdot s)^{2(d-1)/d}. \tag{41}$$

Notice that $a(0) = 0$ for the $d \geq 2$, elucidating the hyperbolic nature of the problem close to the origin. The numerical difficulties of hyperbolic problems can be resolved by using a Lagrangian method: Given a path $s(t)$ in the $(s,t)$ plane. The solution along this path is $T(s(t), t)$ and the total derivative of $T$ with respect to time becomes

$$\frac{dT}{dt} = \frac{\partial T}{\partial s}\frac{ds}{dt} + \frac{\partial T}{\partial t}, \tag{42}$$




which, inserted into Eq. (41) gives

$$\frac{dT}{dt} + \left( v(t) - \frac{ds}{dt} \right) \frac{\partial T}{\partial s} = \frac{\partial}{\partial s} \left( \hat{\lambda} a(s) \frac{\partial T}{\partial s} \right). \tag{43}$$

If we let the path $s(t)$ satisfy $ds/dt = v(t)$, the advection term is eliminated. In fact, the paths $s(t)$ are the characteristics for the hyperbolic equation we obtain for $\hat{\lambda} = 0$. As a result, Eq. (41) can be solved as a system of differential equations:

$$\frac{ds}{dt} = v(t),$$
$$\frac{dT}{dt} = \frac{\partial}{\partial s} \left( \hat{\lambda} a(s) \frac{\partial T}{\partial s} \right).$$

The first equation is an ordinary differential equation, whereas the second one is a heat equation with a space dependent diffusion coefficient. This can be discretized in space by some appropriate finite difference schemes, e.g.

$$\frac{ds_i}{dt} = v(t), \tag{44}$$

$$\frac{dT_i}{dt} = \frac{2\hat{\lambda}}{s_{i+1} - s_{i-1}} \left( a_{i+1/2} \frac{T_{i+1} - T_i}{s_{i+1} - s_i} - a_{i-1/2} \frac{T_i - T_{i-1}}{s_i - s_{i-1}} \right), \tag{45}$$

with $a_{i+1/2} = (a(s_{i+1}) + a(s_i))/2$, and initial values $s_i(0) = s_{i,0}$ and $T_i(0) = T(s_{i,0}, 0)$. The procedure is significantly simplified if $v$ is constant, in which case the characteristics $s(t)$ are just straight lines.

The spacial domain can be extended to $\mathbb{R}$ by defining $a(s) = 0$ for $s < 0$. In this case, we may solve Eq. (41) with the boundary conditions:

$$\lim_{s \to -\infty} T(s,t) = 1, \qquad \lim_{s \to \infty} T(s,t) = 0. \tag{46}$$

When water is injected, $v > 0$ and the temperature of the water at the well is $T(0,t) = 1$. This is realized by choosing $T(s,t_0) = T_0(s)$ whenever $s > 0$ and $T(s,t_0) = 1$ for $s \leq 0$. Here $t_0$ is either the initial time or a switching time, that is whenever $v(t)$ changes from negative to positive (from pumping to injection). The procedure is illustrated for the injection phase in Fig. 3.

In order to be able to resolve a sharp front, the characteristics $s_i(t)$ used in the discretization can be concentrated around it.

**Example 1.** To demonstrate the idea, consider Eq. (40) for $d = 2$, using $\kappa_2 = 1$, $T(0,t) = T_{inj} = 1$ and $T(r,0) = 0$. The exact solution is given by Eq. (39). The transformed system is first solved numerically by the Lagrangian scheme Eq. (44–45). The initial computational domain is $(-S_{int}, S_{int})$ where $S_{int}$ is chosen sufficiently large to avoid any influence from the boundaries. The concentration of characteristics around the front is achieved by using

$$\bar{s}_i = -\frac{1}{2} + \frac{i}{N}, \qquad s_i(t_0) = (\text{sgn}(\bar{s}_i))^{p-1} \bar{s}_i^{\,p} S_{int}, \qquad i = 0, \cdots, N, \tag{47}$$

where $p$ is a positive integer, the higher $p$ the stronger concentration. In our experiments, we have used $p = 3$, $N = 100$ and $S_{int} = 2.4$. The underlying system of ordinary differential equations is solved by a standard solver in MATLAB (ODE15s).

For comparisons, Eq. (40) is also solved by a standard difference scheme with constant stepsize. In this case the advection term is approximated with an upwind scheme, $(\partial T/\partial r)(r_i,t) \approx (T_i(t) - T_{i-1}(t))/\Delta r$. For the diffusion term a central difference scheme is applied. The spatial gridsize is $\Delta r = 0.012$.



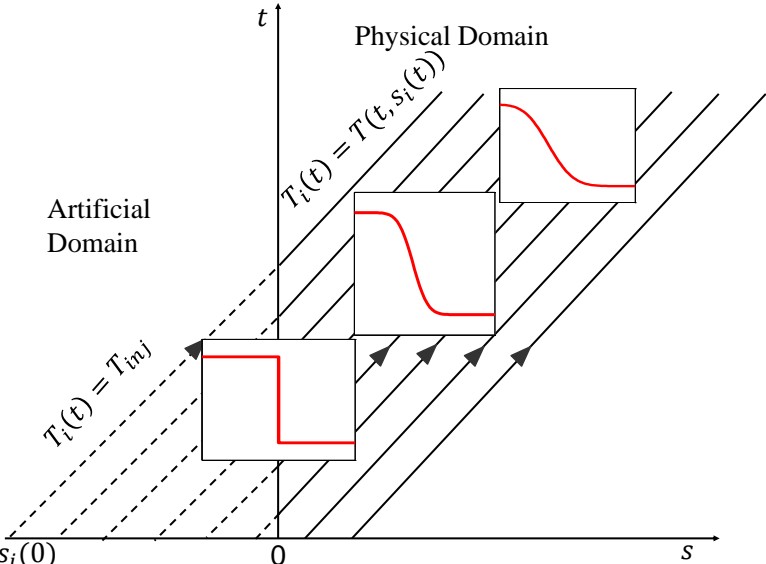

**Figure 3.** The extended domain and the characteristic lines in the injection phase.

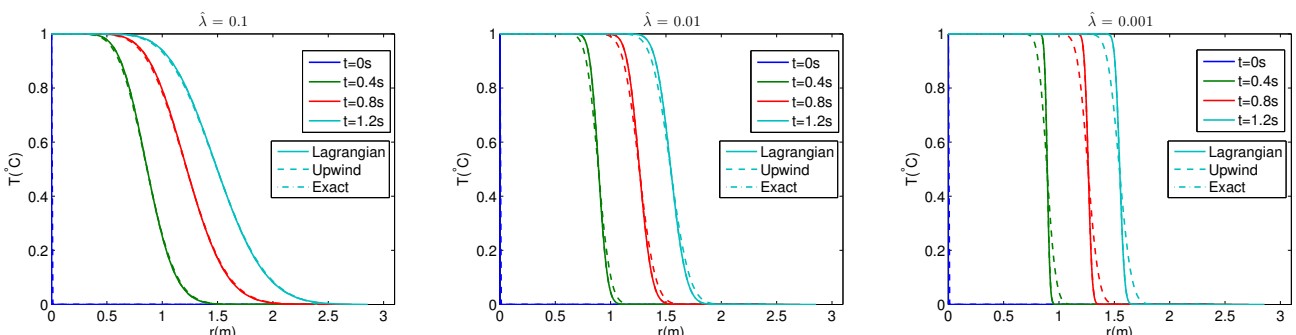

**Figure 4.** Comparison of the numerical solutions based on the Lagrangian method, and a classical upwind scheme of the 2-dimensional radial symmetric case (Eq. 40) for different values of the diffusion parameter $\hat{\lambda}$. The Lagrangian solution overlaps the exact solution for all three values of $\hat{\lambda}$.

The problem is solved for different values of $\hat{\lambda}$, and the results are shown in Fig. 4 together with the exact solutions given by Eq. (39). For $\hat{\lambda} = 0.1$, the diffusion is large and the artificial diffusion of the upwind method is insignificant. For smaller values of $\hat{\lambda}$, the front remains sharp, and the effect of the artificial diffusion of the upwind method becomes quite pronounced. The Lagrangian approach preserves the sharp temperature front.



**The non-equilibrium case**

We now consider the non-equilibrium case Eq. (26–27), which in scaled form becomes

$$\frac{\partial T_s}{\partial t} = \frac{\hat{\lambda}_s}{r^{d-1}}\frac{\partial}{\partial r}\left(r^{d-1}\frac{\partial T_s}{\partial r}\right) + \gamma_s(T_w - T_s),\tag{48}$$

$$\frac{\partial T_w}{\partial t} + v\frac{1}{r^{d-1}}\frac{\partial T_w}{\partial r} = \frac{\hat{\lambda}_w}{r^{d-1}}\frac{\partial}{\partial r}\left(r^{d-1}\frac{\partial T_w}{\partial r}\right) - \gamma_w(T_w - T_s),\tag{49}$$

where

$$\hat{\lambda}_s = \frac{\lambda_s}{\rho_s c_s}\frac{\mathcal{T}}{\mathcal{R}^2}, \qquad \hat{\lambda}_w = \frac{\lambda_w}{\rho_w c_w}\frac{\mathcal{T}}{\mathcal{R}^2}, \qquad \gamma_s = \frac{h}{(1-n)\rho_s c_s}\mathcal{T}, \qquad \gamma_w = \frac{h}{n\rho_w c_w}\mathcal{T}\tag{50}$$

and

$$v = \frac{Q_d}{n}\frac{\mathcal{T}}{\mathcal{R}^d}.\tag{51}$$

Again, for a given time scale $\mathcal{T}$ it is appropriate to choose a spatial scale $\mathcal{R}$ such that $v(t)$ is of the size of 1. The boundary conditions in the injection case ($v(t) > 0$) are

$$T_w(0,t) = T_{inj}, \qquad \frac{\partial T_s}{\partial t}(0,t) = \gamma_s(T_{inj} - T_s(0,t)),\tag{52}$$

(with $T_{inj} = 1$ if the temperature is scaled).

By applying the transformation $s = r^d/d$ and using the Lagrangian approach, we can find the solutions $T_s(s(t),t)$ and $T_w(s(t),t)$ on the characteristics $s(t)$ from

$$\frac{ds}{dt} = u(t),\tag{53}$$

$$\frac{dT_s}{dt} - u(t)\frac{\partial T_s}{\partial s} = \hat{\lambda}_s\frac{\partial}{\partial s}\left(a(s)\frac{\partial T_s}{\partial s}\right) + \gamma_s(T_w - T_s),\tag{54}$$

$$\frac{dT_w}{dt} + (v(t) - u(t))\frac{\partial T_w}{\partial s} = \hat{\lambda}_w\frac{\partial}{\partial s}\left(a(s)\frac{\partial T_w}{\partial s}\right) - \gamma_w(T_w - T_s).\tag{55}$$

where $u(t)$ is the velocity of the temperature front, typically $u(t) = \rho_w c_w n/(\rho c)_m v(t)$. The formulation is used to construct a spatial grid which is dense around the steep solution profile, and moves with it. This is illustrated in the following example.

**Example 2.** Consider the nondimensional equations Eq. (48–49), using $d = 2$ and parameters

$$\hat{\lambda}_w = \hat{\lambda}_s = 10^{-5}, \quad \gamma_w = 1, \quad \gamma_s = 2,\tag{56}$$

and a time dependent flow, $v(t) = \cos(\pi t)$. For $0 < t < 1$, both injection, $0 < t < 0.5$, and pumping, $0.5 < t < 1$, are demonstrated. The equations are solved with the Lagrangian approach, using a central difference approximation for the diffusion terms, and a downwind scheme

$$\frac{\partial T}{\partial s}(s_i,t) \approx \frac{T_{s,i+1} - T_{s,i}}{\Delta s_i}\tag{57}$$





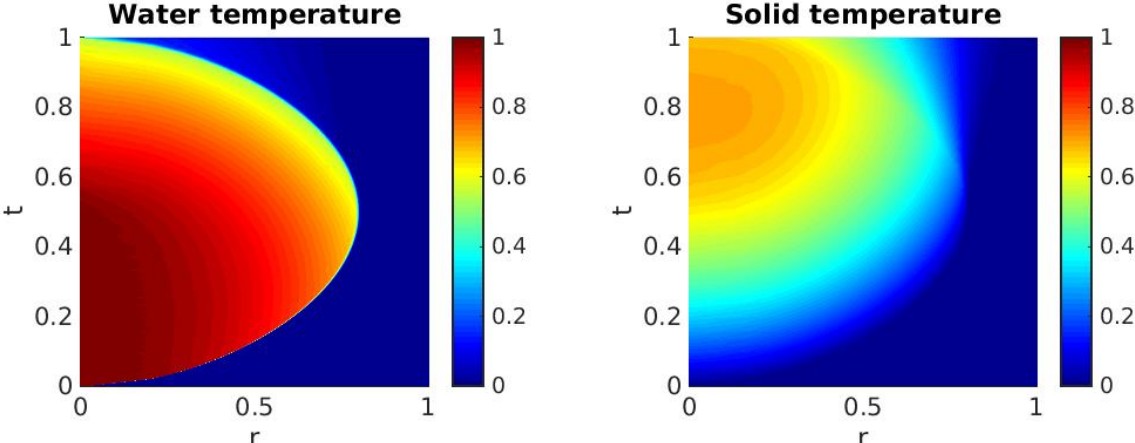

**Figure 5.** Temperature distribution of water (left) and solid (right) in the aquifer as a function of time $t$ and radial distance $r$ for $v(t) = \cos(\pi t)$. Hot water is injected for $t < 0.5$ and pumped for $t > 0.5$.

for the advection term. We have used a spatial stepsize $\Delta s_i$, varying from $1.1 \cdot 10^{-6}$ to $0.14$ and concentrated around the temperature front of the water. The semidiscretized system is solved by MATLABs ODE15s.

The result of the simulation is presented in Fig. 5. We can clearly see how the hot water plume develops with time $t$. The concentration of characteristics moves with the temperature front, which remains sharp. At the same time, there is a heat exchange from water to solid, and the temperature of the water behind the front is reduced at the same time as that of the solid temperature increases. The diffusion is almost negligible most of the time, but will cause a slight smoothing of the front. When $t \to 1$, the front approaches the well with increasing velocity, and the smoothing of the front becomes more pronounced. This example illustrates the numerical challenge, namely the coupling of advective and diffusive physics. Usually, the numerical solutions of such problems are suffering from artifacts, such as numerical diffusion and/or oscillations, but in this case it is possible to suppress the numerical artifacts to negligible levels.

## 5 Case study: Temperature profile near a well in an ATES system

In this section, we consider the temperature propagation around a hot well in an ATES system. The physical and thermal properties of the Gardermoen aquifer were obtained from Goshu and Omre (2003) and Tuttle (1997), and is given in Table 1. The ATES system is typically operating in one of two modi: Injecting hot water during daytime and extracting it during nighttime, resulting in a full operational cycle of 24 hours. Alternatively, the warm water is injected during the summer months, and extracted in the winter, giving a cycle of one year. In reality, a combination of these is used, but in our study, we only consider the first modus, assuming the injection and extraction periods to be of equal lengths, 12 hours.



The Gardermoen aquifer is a delta structure deposited in a glacio-fluvial/glacio-marine environment during the last deglaciation of the Scandinavian crust (approx. 10.000 B.P., Tuttle (1997)). The river discharge and the sediment load from the melting glacier were significant, which explains the wide range of grain sizes of the sediments from boulders ($d_p > 500$mm) at the proximal side of the delta, to fine sand and silt ($d_p \lesssim 1$mm) at the distal side of the delta. The ATES system for this case study

was located in the delta foresets with homogeneous fine sand, but it is interesting to compare the energy efficiency of this ATES with alternative locations. We therefore let the sediments vary form $d_p = 1$mm, which corresponds to a foreset location, to $d_p = 500$mm mimicking a location close to the glacial portals. In that case, the aquifer permeability would have been better, but to keep the experiment as simple as possible, we let the pumping rate and the porosity be the same for all grain sizes. In Table 2 the values of the heat transfer coefficient $h$, Eq. (14), and the time scale towards thermal equilibrium $\Delta t$, Eq. (16), for

different particle size are shown. So we can conclude that within time scales given by the injection/extraction periods, there is almost thermal equilibrium for realistic particle sizes. It is still of interest to see what happens in the initial injection phase, before thermal equilibrium is established, so we will solve Eq. (26–27). In a horizontal confined aquifer we can assume radial symmetry in the vicinity of a well, so $d = 2$. Initial and boundary conditions for the first injection phase is

$$T_w(0,t) = T_{inj}, \qquad\qquad T_w(r,0) = T_0, \qquad\qquad (58)$$

$$\frac{\partial T_s}{\partial t}\Big|_{r=0} = \gamma_s(T_w - T_s), \qquad\qquad T_s(r,0) = T_0. \qquad\qquad (59)$$

The equations are solved by the numerical approach outlined in Section 4.

**Table 1.** Physical and thermal properties of fluid and aquifer for the thermo-hydraulic modelling of the Gardermoen aquifer.

| Property | Symbol | Value |
|---|---|---|
| Porosity | $n$ | 0.1507 |
| Density of fluid | $\rho_w$ | $1000\ \text{kg/m}^3$ |
| Density of aquifer | $\rho_s$ | $2630\ \text{kg/m}^3$ |
| Specific heat of fluid | $c_w$ | 4200 J/kgK |
| Specific heat of solid | $c_s$ | 800 J/kgK |
| Thermal conductivity of fluid | $\lambda_w$ | 0.6 W/mK |
| Thermal conductivity of solid | $\lambda_s$ | 2.0 W/mK |
| Injection/pumping rate | $Q$ | $28\ \text{m}^3/\text{hr}$ |
| Temperature of the injected water | $T_{inj}$ | $30\,^\circ\text{C}$ |
| Aquifer initial temperature | $T_0$ | $4\,^\circ\text{C}$ |
| Aquifer height | $H_0$ | 24.4 m |

**Transient injection phase**

In Fig. 6 we present the temperature profiles for the first few seconds of the injection period. The first row shows the situation for particle size $d_p = 1$mm. Thermal equilibrium happens almost immediately in this case, but the energy transfer still has an





**Table 2.** The heat transfer coefficient (Eq. 14) and the estimated time scale (Eq. 16) towards thermal equilibrium for different particle size.

| $d_p$ | 500mm | 100mm | 10mm | 5mm | 1mm |
|---|---|---|---|---|---|
| $h[\mathrm{W/mK}]$ | 49.3 | $1.2 \times 10^3$ | $1.2 \times 10^5$ | $4.9 \times 10^5$ | $1.2 \times 10^7$ |
| $\Delta t$ | 10.3 hr | 25 min | 15 sec | 3.8 sec | 0.15 sec |

effect in the sense that the temperature front become smoother. Also notice that after 0.15 sec, the solid temperature at the wall has reached to about 2/3 of the water temperature, while the water is almost cooled at the front. This is consistent with the fact that $(1-n)\rho_s c_s / (n \rho_w c_w) \approx 2.8$, thus we expect the water to cool down approximately 3 times as fast as the solid heats up. The lower row gives the same profile for $d_p = 5$mm and $d_p = 10$mm, and as expected, the heat exchange is significantly slower in these cases. As a consequence, the width of the front increases.

Observe the similarities of the top left and the lower right plots in Fig. 6. This is due to the fact that the thermal transfer coefficient $h$ given by Eq. (14) is proportional to $d_p^{-2}$. For the 2-dimensional radial symmetry case, the two solutions may be proved to be identical up to scalings of $t$ and $r$.

**Energy efficiency**

Finally we study the energy recovery from an ATES well based on 12 hours injection and extraction periods. In general, the energy transfer $E$ in the well over a time interval $\tau$ is given by

$$E(\tau) = \int_\tau \rho_w c_w \left( T_{w,0}(t) - T_0 \right) Q(t) \, dt, \tag{60}$$

where $T_{w,0} = T_w(0,t)$ is the water temperature at the well. The efficiency can me measured in terms of the energy recovery factor given by, (Doughty et al., 1982)

$$\theta = \frac{|E(\tau_{extraction})|}{|E(\tau_{injection})|}. \tag{61}$$

Clearly, if the injected water has a constant temperature and the injection rate $Q$ is constant, then $E(\tau_{injection}) = \rho_w c_w (T_{inj} - T_0) Q \tau_{injection}$. During pumping the water temperature at the well will vary depending on the dispersion of temperature in the aquifer, which includes natural heterogeneity and the heat transfer between the liquid and the solid phase. Numerical simulations of the temperature of water and solid at the well over five consecutive cycles are given in Fig. 7. The corresponding recovery rates are shown in the lower right corner of the plot.

We observe that the heat exchange has a significant impact on the efficiency rate for $d_p = 500$mm, otherwise not. We notice that the efficiency recovery rate based on this simplified model corresponds well with the rates achieved for an ATES system in the same aquifer presented in Birhanu et al. (2015b).

The temperature profile over one cycle (injection and pumping) is given for the two extreme cases $d_p = 500$mm and $d_p = 1$mm in Fig. 8.





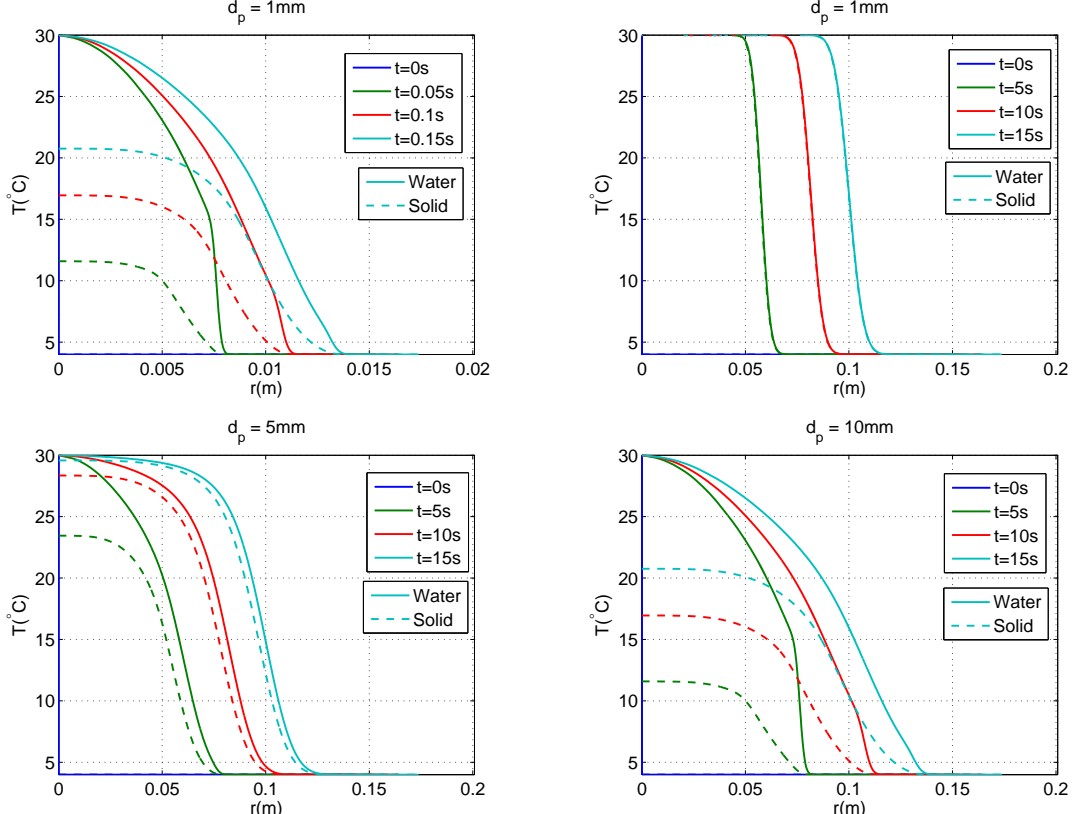

**Figure 6.** The temperature profile of the solid and the water for different values of $d_p$[mm] in 2D radial flow near a well. The solid lines indicate the temperature of the water and the broken lines for solid temperature. The upper row emphasize water and solid temperature profile of the same particle size with different timescale while the bottom row emphasize the water and solid temperature profile of different particle size over the same timescale.

## 6 Conclusion

This paper has briefly reviewed the differential equations for heat transport in water-saturated porous media, and presented numerical and analytical solutions for radially symmetric flow. In particular, a simple similarity solution was obtained for the heat transfer in a 2D horizontal confined aquifer in local fluid/solid thermal equilibrium. For a time varying fluid flow and

5 different fluid and solid temperatures, that is, the non-equilibrium or delayed case, solutions have to be obtained numerically. The numerical algorithms have been based on a semi-discrete Lagrangian formulation.

The numerical models have enabled us to consider the primary purpose of this investigation, namely to calculate the recovery factor of a one-well ATES system with a cyclic repetition of injection and pumping. It has turned out that the performance is dependent on the total length of the cycle relative to the time scale for the heat transfer between fluid and solid. The latter

10 may be linked to the typical grain size $d_p$ as shown in Table 2. For a total cycle of length 24 hours, referring to Fig. 7, the





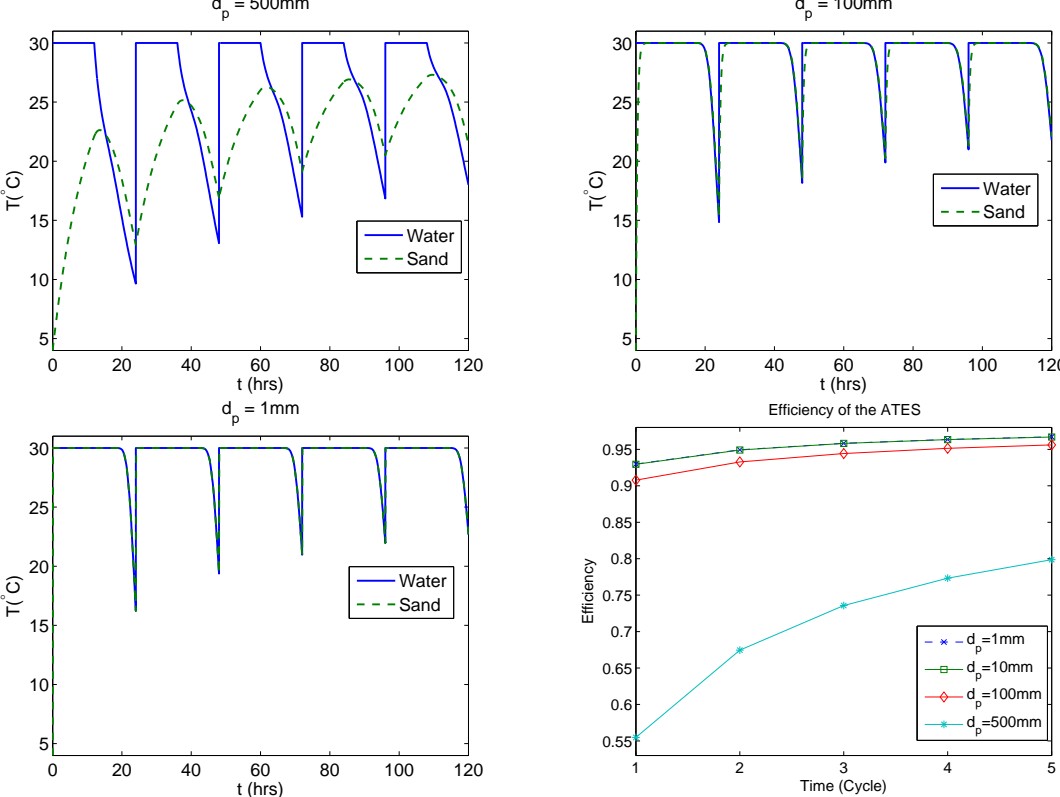

**Figure 7.** The temperature of the water and solid at the well for five consecutive cycles of 24 hours for aquifers of different particle sizes. In the lower right corner, the corresponding recovery rates.

performance is seen to be virtually independent of the grain size as long as $d_p$ is less than about 100mm ($\Delta t \approx 25$minutes), but significantly affected for $d_s = 500$mm ($\Delta t \approx 10$hours). In the latter case, the efficiency is also significantly reduced.

Based on the presented results, the analytic and numerical solutions should provide a consistent tool for the understanding of water and solid temperatures near wells with radial flow.

## 5 7 Acknowledgments

The first author would like to thank Norad's Programme for Master Studies (NOMA) along with the Norwegian Educational Loan Fund (Quota program) for the financial support to conduct this research as a part of his PhD.




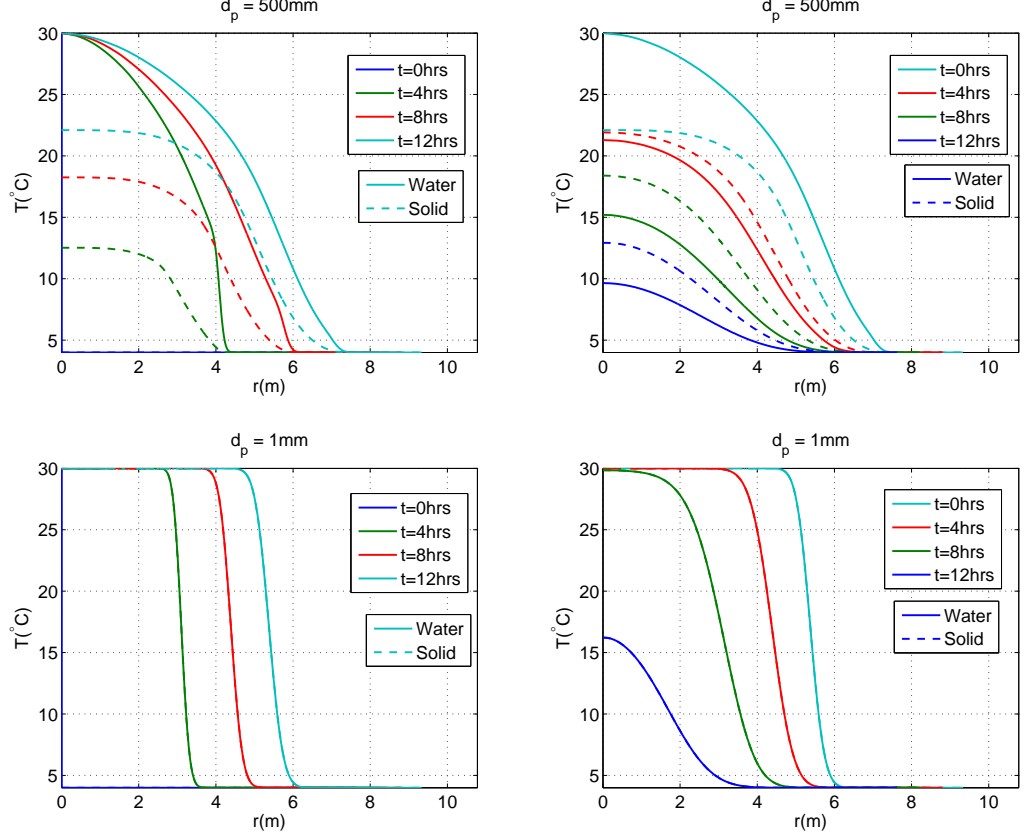

**Figure 8.** Temperature profiles of water (solid lines) and solid (dashed lines) for different particle sizes The left column shows the temperature profiles during injection, the right during pumping.

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
