# Peer review of "Analytical and Numerical Solutions of Radially Symmetric Aquifer Thermal Energy Storage Problems"

_Hydrology and Earth System Sciences, 2017_

## Referee Comment (RC1) · Anonymous Referee #1 · 28 Aug 2017

Review

General comment The authors proposed similarity solutions for equilibrium transport problems in 1-D and 2-D and numerical scheme for non-equilibrium transport problems. However, the model considered in the study is lack of innovation and too simple to represent the real problems. First, the exact analytical solutions for 1-D and 2-D (radial) problems are well known in many literatures. Actually, Yang et al. (2010) have already given a semi-analytical solution for the 3-D problem (radial and vertical directions) with the heat conduction to the overlying and underlying layers in consideration for ATES system. Second, since the exact solution exists, I do not see the necessity

[Figure]

to propose the similarity solution that deviates too much from the exact solution as shown in Figure (1); Third, no field data support the findings of numerical experiments and the importance of non-equilibrium model solved numerically is not demonstrated. Therefore, I recommend to reject this manuscript.

Minor comment (1) For equation (28), the boundary condition at far side should be x->∞ instead of t-> ∞ (2) For equation (31), why propose a similarity solution that is inaccurate compared with the given exact solution? In addition, Figure (1) compares the exact solution with the similarity solution for 1-D problem. How about the accuracy of the similarity solution for 2-D problem? It's better to compare the similarity solution with the exact solution considering the obvious difference in 1-D case. (3) For figures with 2 or more subplots, it might be better to put letters (a,b,c. . .) or numbers (1,2,3..) in front of the subtitle of each subplot to help us understand the sequence of these plots. (4) On page 12 line 22, the meaning of the operation for each time interval is not clear. (5) On page 15 line 61, me->be. (6) On page 17 line 3, provide-> serve as

Reference: Yang, S.-Y., Yeh, H.-D. and Li, K.-Y. (2010), Modelling transient temperature distribution for injecting hot water through a well to an aquifer thermal energy storage system. Geophysical Journal International, 183: 237–251. doi:10.1111/j.1365-246X.2010.04733.x

---

## Referee Comment (RC2) · G. Lu (Referee) · 15 Sep 2017

The manuscript intends to investigate thermal transfer in aquifers through radially symmetric flow system. The underlined goal is what could lead to possible difference in heat transfer in solid grains (matrix) and the pore water in between. It is, however, that pore waters are in small pore spaces well confined by grains that local equilibrium could be reached for water and grains, in a relatively short frame in time and distance.

The flow equation from the start is wrong in equating both sides (weight unit vs. volumetric unit) [in Eq.(1), see detailed specific comment No.5 below]. The thermal flow is not effectively coupled with water flow, because water flow is assumed a constant

throughout the space (rather than considering its variation in a radial space). So the analytical solutions are not solving the flow in Darcy's law [Eq.(1)].

The results presented are mostly for very short time as in several seconds, and thus could be considered to be insignificant due to the limited time duration. Because figures show too fast conductions that thermal effect reaches half a meter in several seconds (Figures 1,2,4), the heat transfer shows the results seemingly wrong.

The figures drawing is poor and the manuscript is hard to follow, especially in notations for the equations. The physical meaning for the parameters is difficult to understand. (See specific comments throughout detailed below.) Considering the deficiency above, this reviewer recommends a rejection for this article to be published in HESS journal.

Specific and detailed comments: 1a. Title The title is vague and misleading.

1b. Abstract Line 4 "....artificial dispersion. There is a good correspondence..." need a better transition

2. p.2 (1 Introduction) Lines 19-20 What is the usefulness regarding this paper?

3. p.2 Line 25 What is the alternative solution?

4. In p.2 There are no objectives here for the research paper.

5. p.2 Eq.(1) The Darcy's law is wrong in formulation. The left side is flow rate in weight per unit area per unit time through a unit area (e.g., kg/mˆ2/s); however, the right side is flow rate in volume per unit time per unit area (e.g., mˆ3/mˆ2/s). They are different by a ratio of density, needing the viscosity $\mu$ to be kinematic viscosity (mˆ2/s).

6. p.3 Line 19 conduction/diffusion of heat What is the difference between conduction and diffusion? Is diffusion significant here?

7. Page 4 Line 11 Eq.(12) P used for heat exchange is confused with pressure p used in Eq.(1).

Interactive
comment

8. Page 4 Line 27 Eq.(17) What is the n here in the formula? What are these terms meant for?

9. Page 5 Line 19 What is the i in the formula?

10. Page 6 Line 10 What is sensitivity analysis here?

11. Page 6 Line 22

12. Page 7 Figure 1 There are no labels for the subplots. The heat diffusion coefficient $\lambda$ 0.1 mˆ2/s values differ so great that these values might become unphysical. (There needs a space before the unit for illustrating $\lambda$.) What does the $\kappa$1 1.0 m/s mean [see Eq.(23)]? This could be too big a number also.

The unit for t (time) is only having the range of 0.1 to 1 s. (Judged from what are shown in Figures 6,7,8, the time t is surely in seconds.) Since the plot is for the temperature distribution over the domain in a very short time, it is unreasonable that the temperature disturbance could be up to 0.5oC in a place 0.5 meter deep inside. Beside, the lines cannot tell apart.

13. Page 9 Figure 2 has a problem similar with that for Figure 1 (see Question 12).

14. Page 9 Line 12 The s has already used for time. Using it again for transformation causes confusion.

15. Page 11 Figure 4 has similar doubt as in Figures 1 and 2.

16. Page 14 Table 1 What are the mK, kgK used in the unit?

17. Table 2 The grain sizes in the table are uncharacteristically large as in 100 mm, 500 mm. In such a large grains, the contact interfaces might just become very small relatively so that heat transfers between grains are not that effective.

18. Figures 6,7,8 These figures miss labels for subplots. The lines are not differential, especially in a black and white print.
19. Page 16 "6. Conclusion" Lines 2-3 "...presented numerical and analytical solutions for radially symmetric flow" The analytical solutions presented are actually not for flow. The flow equations were not solved for flow velocity. This reviewer believes only heat transfer equations were solved or discussed, with the assumption that the specific flow q (in Eq.(1)) is constant (See text in Page 5 Lines 17-20, Eqs. 22, 23, 24). (In a radial flow, q should be a variable.)

20. Page 17 Lines 3-4 The results from the analytical solutions are shown at the time of only several seconds. The thermal response could be too dramatic to be realistic in such a short several seconds of time, given that the characteristic time scale for thermal conductions is much larger.

---

## Referee Comment (RC3) · Anonymous Referee #3 · 30 Sep 2017

The authors used analytical and numerical methods to investigate heat transport in saturated porous media. An approximate solution for 1D case and both analytical and numerical solutions for 2D case are presented. Case studies are also given to show temperature profile near a well in an ATES system.

The manuscript is poorly written. The structure of this manuscript is difficult to follow. The authors do not show clearly what are done by this study.

The authors did not cite any paper from this journal. The reviewer doubts where this study fits the aim and scope of this journal or not. There should be some of the papers in the journal have done research work similar to this study. The authors should at

least cite a few most related papers.

The symbolic system in the manuscript is chaotic, the formula are logically unconnected and adds an additional reading burden.

The time scale in the models and solutions are not reasonable. Many of them are less than 1 s. Such times are not comparable to real systems. The authors should clarify this point.

There are many unphysical parameters in the manuscript. For example, a 500 mm grain size in Table 2 and a k1=1 m/s in Figure 1. Both of them are too large. The authors should use reasonable parameter values in order to make the results useful.

Specific comments: (1) Equation (31): As the exact solution already exists, there is no need to derive an inaccurate approximate solution. Comparing equations (31) and (32), one can find that equation (31) is just the first term of equation (32). As can be seen from Figure 1, equation (31) is inaccurate and it is not a good approximation.

(2) Figures 1, 2, 4, 5: The temperature T ranges from 0 to 1 °C. Such temperatures are too low for a thermal energy storage problem. The authors should explain why the temperatures are so low. These temperatures are not reasonable and unphysical.

(3) Since an exact solution equation (39) is already exist, there is no need to present the numerical solutions. If the Lagrangian solution is used to examine the correctness of the exact solution, then why the authors also present the upwind solution here?

(4) Page 5 Line 9: The author use "may". It increases the uncertainty of the model. Whether the two are equal also need to be discussed.

(5) Page 5 Line 9: in "q = qi" formula the variable i does not have a proper description, it is puzzling.

(6) Page 12 Line 22: The unit for t should be given. Is it in seconds? If yes, then t in Figure 5 is also in seconds. In Figure 5, the authors stated that "Hot water is injected

for t<0.5 and pumped for t>0.5". The hot water injection is only 0.5 s. Such a short time is unphysical. Why the author did not use time scales in the analytical and numerical solutions comparable to real cases? For example, 12 hours used in Section 5. Or the analytical and numerical solutions are incorrect and cannot be used for real cases?

(7) Page 15 Line 13: "can me" should be "can be"

(8) Page 15 Lin 24: dp=1 mm is a reasonable value, why the authors use it as an extreme case?

(9) Figure 6: The upper right corner of the sub-figure is puzzling. Why it is so difficult from the other sub-figures?

(10) The conclusion section should be rewritten. The authors should put main conclusions here.

In conclusion, the reviewer suggest rejecting this manuscript.

―――――――――――――――――